# Distinctly Different Morphologies of Bimetallic Au-Ag Nanostructures and Their Application in Submicromolar SERS-Detection of Free Base Porphyrin

**DOI:** 10.3390/nano11092185

**Published:** 2021-08-26

**Authors:** Iveta Vilímová, Karolína Šišková

**Affiliations:** Department of Experimental Physics, Faculty of Science, Palacký University in Olomouc, Tř. 17. Listopadu 1192/12, 77146 Olomouc, Czech Republic; vilimovai4@gmail.com

**Keywords:** bimetallic nanoparticles, silver-gold particles, seeded-growth, reactant order, nanoparticle morphology, TMPyP, free base porphyrin, non-metalated porphyrin, SERS

## Abstract

Core-shell Au-Ag nanostructures (Au-AgNSs) are prepared by a seed-meditated growth, i.e., by a two-step process. The synthetic parameters greatly influence the morphologies of the final bimetallic Au-AgNSs, their stability and application potential as surface-enhanced Raman scattering (SERS) substrates. Direct comparison of several types of Au NPs possessing different surface species and serving as seeds in Au-AgNSs synthesis is the main objective of this paper. Borohydride-reduced (with varying stages of borohydride hydrolysis) and citrate-reduced Au NPs were prepared and used as seeds in Au-AgNSs generation. The order of reactants in seed-mediated growth procedure represents another key factor influencing the final Au-AgNSs characteristics. Electronic absorption spectra, dynamic light scattering, zeta potential measurements, energy dispersive spectroscopy and transmission electron microscopy were employed for Au-AgNSs characterization. Subsequently, possibilities and limitations of SERS-detection of unperturbed cationic porphyrin, 5,10,15,20-tetrakis(1-methyl-4-pyridyl)21H,23H-porphine (TMPyP), were investigated by using these Au-AgNSs. Only the free base (unperturbed) SERS spectral form of TMPyP is detected in all types of Au-AgNSs. It reports about a well-developed envelope of organic molecules around each Au-AgNSs which prevents metalation from occuring. TMPyP, attached via ionic interaction, was successfully detected in 10 nM concentration due to Au-AgNSs.

## 1. Introduction

Among metal nanostructures (NSs), gold and silver NSs are the most commonly used candidates in optical, catalytic, biomedical applications [1,2] due to their unique physico-chemical properties. The most intriguing property of Au and Ag NSs is their localized surface plasmon resonance (LSPR) [3,4,5,6,7] which is for these noble metal NSs conveniently located in the visible (vis) and near infrared (NIR) regions. The LSPR occurs due to d-d band transitions and makes Au and Ag NSs particularly suitable for biomedical applications [7]. Exposing Au or Ag NSs to light leads to many LSPR-related processes, including Mie extinction (absorption and scattering), surface-enhanced Raman scattering (SERS) and photothermal effect [1].

The combination of Au and Ag NSs in core-shell arrangements is preferentially used and investigated because different shapes and morphologies of the final NSs can be created, not only spherical, but also triangular, cubic, dumbbells, bipyramids, pentagonal rods, wires, hexagonal platelike, octahedral, and multiple-twinned decahedral structures in refs [1,2,7,8,9,10,11,12]. The combination of these two noble metals prepared as core-shell structures revealed improved properties, namely in LSPR-based and SERS-based spectroscopic detection of different species (e.g., As(III) [13], Cr(VI) [14] etc.), in comparison to pure Ag and/or Au nanostructures (e.g., [15,16]). 

Owing to the similarity of Au and Ag lattice constants (0.408 and 0.409 nm, respectively) [17], these two noble metals can be easily combined within one nanostructure. It should be noted that the co-reduction of HAuCl_4_ and AgNO_3_ usually results in nanoalloys (e.g., [18,19,20]), although under specific conditions (usage of pomegranate fruit juice as reducing agent and Au(III):Ag(I) molar ratio of 1:1) [9] Au-Ag core-shell NPs were generated as well. 

Generally, the reproducible preparation of Au-Ag core-shell NSs requires a two-step process; seed-mediated growth and weak reducing agents, such as for instance L-ascorbic acid, are used. Ascorbic acid (Asc), also known as Vitamin C, is considered a relatively weak reductant under non-alkaline conditions [21]. This could benefit the synthesis of metal NSs via seed-mediated growth because the self-nucleation of newly formed metal atoms is suppressed [22]. Asc forms three different species depending on the solution pH: ascorbic acid, ascorbate, and diascorbate. For all three forms, the mechanism involved in the syntheses of noble-metal NSs are similar to each other in terms of electron transfer and the oxidized product formed in the redox reaction (dehydroascorbate) [21]. Interestingly, the seed-mediated growth process of Au(core)-Ag(shell)NSs was employed for the quantitative LSPR-based detection of vitamin C concentration [23]. 

Ding et Zhu [24] stated that a different amount of Au seeds (generated from HAuCl_4_, CTAB, NaBH_4_) resulted in different size of Au nanocubes (prepared in the growth solution containing Asc, CTAB, HAuCl_4_) which served as seeds in the next step of Au-Ag nanocubes preparation (i.e., addition of CTAB, AgNO_3_, Asc, NaOH). The thickness of silver shell was fine-tuned by changing the volume of gold nanocubes and silver nitrate [24]. 

Similarly, Pei and coworkers [25] reported that by varying the concentration of AgNO_3_ and amount of Au seeds (synthesized by using sodium citrate and HAuCl_4_), the size of Au-Ag nanospheres (arising from the seed mediated growth by adding Asc and AgNO_3_) can be controlled. 

From the above mentioned examples, it is obvious that detailed experimental conditions are crucial in the generation of Au-Ag core-shell NSs of different morphologies and varying shell-thicknesses. Until now, several parameters have been thoroughly investigated: various types of additional capping agents on seeds and changes in their concentrations, varying concentrations of Au seeds and AgNO_3_, changes in reaction times and stirring rates [11,12,15,20,23,24,25,26,27,28,29,30,31,32,33]. 

Concerning the order of reactants in seed-mediated growth using Asc, very limited attention has been paid to this factor in the literature so far [27,28,30]. Zheng and co-workers [30] demonstrated that the order of reactants played a key role in the formation of Au-AgNSs and their LSPR occurrence. However, their [30] reaction mixture contained CTAB (besides HAuCl_4_) which formed micelles upon addition of Asc. 

Due to the mean size of the as-prepared Au-Ag core-shell NSs (tens of nanometers), they can be exploited in the detection of low concentrations of a model molecule by SERS [34]. Frequently, rhodamine 6G (R6G) (e.g., [11,24,26,29,35]) and/or crystal violet (CV) (e.g., [8]) were used by many authors as model molecules evaluating SERS-efficiency of the generated Au-AgNSs. However, we are systematically using porphyrins as SERS-reporter biomolecules because they can better evaluate the surface features of NSs, such as metalation predispositions (i.e., the predispositions for incorporation of a metal ion into a porphyrin core), oxidation state of Ag ions at the surface of NSs, etc. [36,37,38,39,40,41]. Based on previous studies, tetracationic TMPyP (5,10,15,20-tetrakis(1-methyl-4-pyridyl)21H,23H-porphine) can interact with Ag(I) and/or Ag(0) surfaces, revealing metalation markers in slightly different positions leading thus to different SERS-spectral patterns [40,41,42,43]. The metalation of a free base porphyrin molecule represents an undesirable effect from the viewpoint of the unperturbed free base porphyrin SERS detection in bioanalytical applications [44,45]. There were several attempts to prevent the metalation (i) by employing spacers between TMPyP and Ag surface [37] and/or (ii) by changing the envelope around Ag NPs via different synthetic procedure [38]. The role of citrates on the surface of AgNPs serving as a pre-orienting matrix for porphyrin molecules (not only TMPyP, but also other types of porphyrins) was found out by us in [39].

In this paper, we are dealing with the creation of Au (core)—Ag (shell) NSs bearing resemblance to flowers of different morphologies, which is achieved simply by changing the Au seed type. In general, flower-like nanoparticles possess nanometer-sized intersticies on the surface which results in SERS enhancement. In our case, the Au-Ag core-shell arrangement is the most convenient one for several reasons: (i) AgNSs represent better SERS enhancers than AuNSs [15,26]; (ii) reproducibility and monodispersity of AuNSs, serving as seeds (in the next step of the synthesis), are much better than that of Ag. This is due to the fact that gold NSs can be easily made, measured, and modelled which is stated as the 3M’s principle in the literature [5]. (iii) Galvanic replacement (which takes place in the reverse structure, i.e., Ag-Au core-shell—e.g., [46]) is avoided. 

In the present study, two different, but well-known types of AuNPs were synthesized and they served as seeds in the next step of the final Au-AgNSs preparation: borohydride- and citrate-reduced AuNPs, labelled as AuBhr and AuCitr, respectively. These two were intentionally chosen for their general frequent usage, but nobody has compared them directly yet when serving as seeds in Au-AgNSs generation. A few facts about each reducing agent, of which residues are present at Au seed surface, should be reminded here.

Sodium borohydride is a strong reductant, often used in metal nanoclusters and NPs formation because it reduces a salt precursor within a very short time. The fast formation of plenty tiny metal nanocrystals leads to growth via attachment [47] and thus, it limits the utility of NaBH_4_ in the shape-controlled synthesis [21]. Mechanistically, the important step during the redox process using NaBH_4_ is its hydrolysis to B(OH)_4_^−^ and consequently, surface chemistry on noble-metal nanostructures in the early stages of their formation [48,49]. Of course, the redox potential of NaBH_4_ strongly depends on the pH of the reaction solution; it can vary from 0.48 to 1.24 V when the pH is changed from neutral to alkaline [21]. As the time passes, polyborates are formed from B(OH)_4_^−^ [50,51,52]. 

Sodium citrate is also a well-established reductant for the syntheses of noble-metal nanostructures due to its abundance and versatility to serve as both the reductant and stabilizer [12,21,39]. It can be found in four different forms as a function of pH and by increasing the pH value, an increased electron density available for donation during a redox process is achieved which leads to a stronger reducing capability [53]. Mechanistically, there are two reductants involved: (i) sodium citrate itself and (ii) acetone generated during the redox reaction [21].

According to our best knowledge, the direct comparison of the impact of Au seeds, differing by their surface species stemming solely from syntheses (i.e., poly/borates, borates, hydroxyborates and citrate-residues), on the final morphology of Au-Ag core-shell NSs has never been addressed so far. Therefore, it is the first aim of this study.

As the second main goal of this work, our attention is devoted to the order of reactants in the second step of seed-mediated growth of the final Au-AgNSs. Usually, the second step of Au-AgNSs generation includes the preparation of a mixture of AgNO_3_ and Au seeds (whatever type), followed by the addition of Asc solution, i.e., Asc is added at the end of the seed-mediated growth procedure (e.g., [11,15,20,23,24,28,32,33]). Inspired by the work of Polte et al. [54,55], where the principles of metal NPs growth were discussed based on the concept of colloidal stability, we investigated the influence of the reverse order of reactants (i.e., Asc prior to AgNO_3_).

Finally, selected Au-AgNSs were employed in SERS detection of the model porphyrin TMPyP in its free base form. Thus, here described and investigated synthetic procedure of Ag shell generated on Au seeds, represents a new possibility of synthetic approaches to reach unperturbed (non-metalated) porphyrin SERS spectra.

## 2. Materials and Methods

### 2.1. Chemicals

Silver nitrate (AgNO_3_, 99.999% purity, abbreviated as “Ag+” in sample code), sodium borohydride (NaBH_4_, >98% purity), tetrachloroauric acid (HAuCl_4_, >99% purity), L-ascorbic acid (≥99.0% purity, abbreviated as “Asc” in sample code), 5,10,15,20-tetrakis(1-methyl-4-pyridyl)21H,23H-porphine molecules (TMPyP, 97% dye content) were purchased from Sigma-Aldrich (St. Louis, MO, USA) and used as received (without any further purification).

All stock solutions were prepared with deionized water (purification system Milli-Q, Millipore Corp., Bedford, MA, USA). All glassware was cleaned with aqua regia acid (the mixture of nitric acid and hydrochloric acid in the ratio of 1:3) and copious amounts of deionized water prior to its usage.

### 2.2. Nomenclature

For easier comprehension and better orientation, a naming convention was developed for our seed solutions and final colloidal NP systems. Seed solutions are either freshly prepared (denoted by using “fr” in their code) or aged for 7 months (without any additional code)—Table 1. Different types of AuNPs serving as seeds are used: either borohydride-reduced (AuBhr), or citrate-reduced (AuCitr). In the case of freshly prepared AuNP seeds using NaBH_4_ as a strong reducing agent, we distinguish between slightly hydrolyzed (labelled as “BH”), and hydrolyzed (“BOH”) borohydride (i.e., aged for 2 h in an ice bath). The order of reactants in the second step of our seeded growth method is also reflected in the sample code as can be read in Table 1.

### 2.3. Syntheses of the AuNP Serving as Seeds

(a)synthesis of AuBH-fr

The seed solution is synthesized by mixing 5 mL of freshly prepared NaBH_4_ solution (4.48 mM, submerged in an ice bath) with 5 mL of 0.94 mM HAuCl_4_ aqueous solution, and 10 mL of deionized water. The solution was then stirred at the rate of 1150 rpm for 5 min. Subsequently, 0.5 mL of this AuNP seed solution was used for the synthesis of AuBH-fr_Ag+_Asc.

(b)synthesis of AuBOH-fr

The seed solution is synthesized by the same procedure as in (a), however, approx. 2-h-aged NaBH_4_ solution (4.48 mM, submerged all the time in an ice bath) was used. Subsequently, 0.5 mL of this AuNP seed solution was used for the synthesis of AuBOH-fr_Ag+_Asc.

(c)synthesis of AuBhr (borohydride-reduced AuNPs aged for 7 months)

AuBhr seed solution was prepared according to the procedure published in [56] with small modifications—instead of reduction of Ag(I) salt, reduction of Au(III) salt is performed. Briefly, 3.43 mg NaBH_4_ was dissolved in 75 mL of deionized water submerged in an ice bath. The mixture was stirred at the rate of 1000 rpm while adding drop-wise 9 mL of 2.2 mM HAuCl_4_ in the middle of the stirring vortex. The mixture was withdrawn from an ice bath after 6 min and subsequently stirred for 1 h until reaching room temperature. The AuBhr colloid was then stored in dark at room temperature for 7 months (such long time of ageing was induced by the limited access into the lab due to the COVID-pandemic) and used for the preparation of AuBhr_Ag+_Asc and AuBhr_Asc_Ag+ systems.

(d)synthesis of AuCitr-fr and AuCitr (citrate-reduced AuNPs freshly prepared and aged for 7 months, respectively)

It is a slightly modified Lee-Meisel method [57]. Briefly, AuCitr seed solution was obtained by bringing 200 mL of 0.2% sodium citrate solution to its boiling point in 500-mL Erlenmeyer flask and adding HAuCl_4_ aqueous solution prepared by mixing 20.52 mg of HAuCl_4_ in 10.26 mL of deionized water. The mixture was kept at boiling point by IKA C-MAG HS7 magnetic stirrer with ceramic heating plate for 1 h. Afterwards, the mixture was stirred for additional one hour until reaching room temperature. Freshly prepared AuCitr particles were used as seeds for AuCitr-fr_Ag+_Asc synthesis. The AuCitr colloid stored in dark at room temperature for 7 months (as mentioned above, due to COVID-pandemic) was then employed for the preparation of AuCitr_Ag+_Asc and AuCitr_Asc_Ag+ systems.

### 2.4. Syntheses of the Final Colloidal Au-AgNSs

Concentrations and volumes of AgNO_3_ and Asc were kept constant, as well as the same volume of Au seeds exploited in all preparations of Au-AgNSs as follows:(i)AuBH-fr_Ag+_Asc: 0.5 mL of AuBH-fr was added to 12.5 mL of 0.2 mM AgNO_3_ aqueous solution, followed by 250 µL of 10 mM ascorbic acid aqueous solution. The final colloidal Au-AgNSs were stirred at the rate of 1350 rpm for 10 s, then at the rate of 850 rpm for additional 25 min.(ii)AuBOH-fr_Ag+_Asc: the same synthetic procedure as (i), however, 0.5 mL of AuBOH-fr was employed. The stirring and timing were the same as in (i).(iii)AuBhr_Ag+_Asc: the same synthetic procedure as (i), however, 0.5 mL of AuBhr colloid was added. The stirring and timing were the same as in (i).(iv)AuCitr-fr_Ag+_Asc: the synthetic procedure was the same as in (i), however, 0.5 mL of AuCitr-fr was introduced. The stirring and timing were the same as in (i).(v)AuCitr_Ag+_Asc: the synthetic procedure was the same as in (i), however, 0.5 mL of AuCitr (i.e., aged AuCitr seeds) was employed. The stirring and timing were the same as in (i).(vi)AuBhr_Asc_Ag+: the procedure of this Au-AgNSs preparation was very similar to that described in (iii) with the only exception: a reversed order of reactants was used during the second step. Briefly, 0.5 mL of the seed solution AuBhr (aged for 7 months) was added to 250 µL of 10 mM ascorbic acid aqueous solution, followed by the addition of 12.5 mL of 0.2 mM AgNO_3_ aqueous solution. The stirring and timing were the same as in (i).(vii)AuCitr_Asc_Ag+: the synthetic procedure was the same as in (vi), however, 0.5 mL of AuCitr (instead of AuBhr) was employed.

Based on the assumptions that generated Au seeds are spherical with the mean diameter of approx. 20 nm (and/or 50 nm) and all Au(III) was consumed into Au seeds, the minimal concentration of Ag(I) which covers the whole surface of Au seeds (monolayer Ag(I) coverage, i.e., Ag atoms placed next to each other on Au surfaces) can be calculated (details in Appendix A). Taking into account the concentrations and volumes of Au(III) used in our experiments (AuBhr and AuCitr serving as seeds), the amount of the monolayer Ag(I) coverage is around 9.2 × 10^−9^ mol for 20 nm-size Au seed particles (while 3.7 × 10^−9^ mol Ag(I) for 50 nm-size Au seed particles—Appendix A). Hence, in both cases (AuBhr, AuCitr), the employed Ag(I) amount exceeds the monolayer Ag(I) value by more than two orders of magnitude.

All syntheses were prepared in triplicates by two different experimentalists to exclude any discrepancy and verify the reproducibility.

### 2.5. Methods

UV-Vis spectra of freshly prepared and aged AuNP seeds, as well as final colloidal Au-AgNSs were recorded on spectrometer Specord PLUS 250 (ChromSpec, Analytic Jena, Jena, Germany), in the range between 190–1100 nm. Quartz cuvettes with 2.5 mL of a particular colloidal sample were used for the measurements. Deionized water in another quartz cuvette served as a reference.

Extinction spectra for kinetics measurements were recorded on the same UV-Vis spectrometer in the range between 190–900 nm. Quartz cuvette was filled with 750 µL of the sample taken from the reaction mixture in a particular time interval elapsed from the addition of the last reactant of seed-mediated growth of Au-AgNSs (i.e., after Asc or AgNO_3_).

Transmission electron microscopic (TEM) imaging were performed by using Jeol 2010F transmission electron microscope (Jeol USA, Inc., Peabody, MA, USA), equipped with a LaB6 cathode (accelerating voltage 80 kV–200 kV; CCD camera KEENview G2 (ResAlta Research Technologies, Golden, CO, USA). A 3-µL drop of a particular colloidal sample was deposited onto a carbon-coated copper grid. Grids were allowed to dry at room temperature in a Petri dish covered by its lid for at least one day before performing TEM measurements.

Energy dispersive spectra (EDS) of selected samples were recorded on Oxford x-MAX 80T (SSD) (Oxford Instruments, Oxford, UK).

Dynamic light scattering (DLS) was employed to determine the size distribution of AuNP seeds and of final colloidal Au-AgNSs directly in solution. Zetasizer Nano Series (Malvern Instruments Ltd., Malvern, UK) using the laser beam of 633 nm wavelength, operating in the scattering angle of 173° was employed. The measurement was performed at 25 °C in a disposable polystyrene cuvette filled with 1 mL of a particular colloidal sample, which was diluted with deionized water at 1:10 *v*/*v* ratio. The measurement was set as “multiple narrow mode” resolution and automated number of “runs” during the measurement. A mean result of three measurements was recorded. Particle size distribution according to intensity is considered in our results and discussion section because it is the directly measured size distribution. Besides particle size distribution according to intensity, the software of Zetasizer Nano Series (Malvern Instruments Ltd, Malvern, UK) enables estimation of the size distribution according to number and/or volume. However, both latter size distributions are calculated with some assumptions which are not always valid in our systems and, thus, cannot be accepted. Therefore, we show and discuss only the size distribution based on intensity changes.

The same Zetasizer Nano Series (Malvern Instruments Ltd, Malvern, UK) was used for measuring zeta-potential of colloidal AuNP seeds and Au-AgNSs. The 0.75 mL of a particular sample (no dilution) was placed into a disposable zeta-potential cell and a mean result of three measurements was recorded.

pH values of the AuNP seeds and final colloidal Au-AgNSs were measured by using a standard laboratory pH meter inoLab (type 7110, Xylem Analytics, GmbH & Co, Mainz, Germany).

Surface-enhanced Raman scattering (SERS) spectra were recorded on ProRaman-L spectrometer (TSI, Shoreview, MN, USA) equipped with diode laser of 785 nm wavelength (adjustable optical power output from zero to 300 mW, electronic laser shutter control); high throughput fiber optics probe (O.D. > 8 at laser wavelength); high sensitivity, ultra-low noise CCD spectrograph for 785 nm excitation (thermoelectrically cooled CCD detector to −60 °C); and RamanReader^®^-L7B1 instrument control and data collection software. The SERS spectra of Au seeds and/or Au-AgNSs without and with TMPyP (1 × 10^−8^ M in the final SERS systems, as well as 1 × 10^−6^ M in the case of Au seeds) were measured in 1-cm quartz cuvettes (3/Q(10, ChromSpec, spol. s.r.o., Brno, Czech Republic) by using 300 mW laser power, integration time of 2 s and 60 acquisitions. The SERS spectra were recorded in the spectral range from 100 to 3300 cm^−1^, average spectral resolution of approx. 7 cm^−1^ (pixel resolution of approx. 1.8 cm^−1^ per pixel). Any smoothing and/or corrections of the recorded SERS spectra were omitted, automatic baseline correction was performed.

## 3. Results and Discussion

### 3.1. Direct Impact of Au Seed Type on Morphologies of Final Au-AgNSs

The effect of Au seed types on the morphologies of the final Au-AgNSs was investigated using TEM imaging. The characteristic TEM images of all Au seeds used in the present study are shown in Appendix A. Since no polymers and/or large organic molecules (serving as capping agents) are used in our syntheses of Au-AgNSs, no organic layer around the final metallic NSs is detected using TEM. The contrast in TEM images of Au-AgNSs thus mirrors the presence of AuNPs, darker, and AgNPs, lighter. EDS (energy dispersive spectroscopy) was also performed in selected cases of the final Au-AgNSs in order to give evidence about the simultaneous Au and Ag presence (Appendix A).

#### 3.1.1. Au-AgNSs Stemming from Au Seeds Prepared by Reduction Induced by Borohydride

In order to demonstrate the influence of borohydride hydrolysis (and related changes in surface chemistry) during the formation of Au seeds, we investigated the effect of the degree of NaBH_4_ hydrolysis on the final Au-AgNSs morphology. Indeed, freshly prepared (denoted as BH) and/or two-hours aged borohydride (however, stored in an ice-bath so that the hydrolysis is slowed down and reduction strength is retained; denoted as BOH) were used for AuNPs generation. AuNPs were, immediately after their preparation (when a characteristic red color of solution appeared), exploited as seeds (therefore denoted as “fr”) in the next step of Au-AgNSs formation. For the sake of a direct comparison, aged AuBhr serving as seeds (denoted simply as AuBhr) were also used for Au-AgNSs generation.

TEM imaging of the final Au-AgNSs prepared by using Au seeds generated via fresh vs. few-hours-aged borohydride (AuBH-fr_Ag+_Asc and AuBOH-fr_Ag+_Asc, respectively) revealed completely different morphologies (Figure 1A–D, respectively) in comparison to those observed for aged Au seeds, i.e., AuBhr_Ag+_Asc (Figure 1E,F). AuBH-fr_Ag+_Asc manifested itself by Au cores surrounded by AgNPs (of much smaller sizes than Au seeds) which are directly attached to the surface of Au cores (Figure 1A,B). This resembles flower-like particles with many nanometer-sized interstices and SERS enhancement can be envisaged. In the case of AuBOH-fr_Ag+_Asc, a mixture of Au and Ag NPs relatively randomly dispersed was visualized (Figure 1C,D). In TEM images of AuBhr_Ag+_Asc (Figure 1E,F), each AuNP is surrounded by many small NPs which are supposed to be AgNPs. This assumption is made on the basis of a direct comparison with TEM images of particular Au seeds (Appendix A).

Interestingly, Au seeds generated by borohydride of different stage of hydrolysis are of different sizes and shapes: bigger and more asymmetrical in AuBH-fr than in AuBOH-fr (compare Figure 1A–D, as well as Appendix A) although the time elapsed for their generation was the same. It confirmed the differences in reduction strength of freshly prepared vs. few-hours-aged borohydride; the latter being of a weaker reduction strength. Consequently, varying surface reactions are taking place on noble metal surfaces in the early stages of their formation. Similarly, this was demonstrated in the case of AgNPs in [48].

Furthermore, taking into account TEM results in Figure 1, it is thus a simple experimental proof that the degree of borohydride hydrolysis in the course of Au seeds generation has a strong impact not only on the size and shape of Au seeds, but also on the final Au-AgNSs morphology.

#### 3.1.2. Au-AgNSs Stemming from Au Seeds Prepared by Reduction Induced by Citrate

By changing the type of a reduction agent in Au seeds formation from borohydride to citrate, distinctly different morphologies were obtained. While AuNPs enveloped by a bunch of tiny AgNPs in the case of borohydride-reduced Au seeds were observed (Figure 1); a compact Ag layer around AuNPs was detected in the case of Au-AgNSs when citrate-reduced Au seeds were employed, i.e., either AuCitr-fr, or AuCitr (Figure 2). The latter arrangement is much more pronounced in the literature regardless of the (i) conditions of Au seed preparation: the same synthesis as in our case (e.g., [58]), whereas photochemically prepared Au seeds [59]; (ii) shape of Au seeds (e.g., [10]).

#### 3.1.3. Reason for Different Morphologies of Au-AgNSs

It can be summed up that three different types of morphologies, schematically depicted in Figure 1, were obtained simply by changing Au seeds. While type I represents a mixture of Au and Ag NPs, type II resembles flower-like particles with nanometer-sized interstices, and type III is characterized by a consistent layer of Ag around Au (Figure 1). The reason for morphological differences of the final Au-AgNSs in the present study lies most probably in a completely different surface chemistry and species which are present at each type of Au surface. In AuBhr seeds, borates and polyborates (in aged systems) can be encountered, whereas residues of citrates and acetone in AuCitr seeds can be found. Citrates are known as chelating agents of silver cations [21,39]. Obviously, residues of citrates and/or acetone enable creation of a compact Ag layer around almost each Au seed when seeded-growth procedure of Au-AgNSs is performed. Therefore, it can be deduced that the creation of type III is caused by the chelating effect of citrate ions on silver cations.

Morphologies of type II (Figure 1), observed when AuBH-fr seeds employed (Figure 1A,B), can be caused by the presence of still reactive borohydride on Au seeds. Indeed, it leads to the direct reduction of Ag+ in the very close vicinity of freshly prepared Au seeds.

On the contrary, morphologies of type I (Figure 1), which are characteristic for AuBhr_Ag_Asc and AuBOH-fr_Ag_Asc (Figure 1C–F), are most probably induced by the presence of either polyborates, or borates which hinder the direct deposition of Ag+ on Au seed surfaces (i.e., poly/borates are not chelating agents of silver cations).

### 3.2. Other Characteristics of Au-AgNSs Prepared by Using AuBhr or AuCitr Seeds

#### 3.2.1. Sizes of Au-AgNSs Determined by DLS and Values Discussed in Comparison to TEM

DLS measurements of the final Au-AgNSs were performed in their “native” environment, i.e., in aqueous solutions, to confirm sizes and aggregation state of NSs which were visualized by TEM imaging. It is generally accepted that during TEM samples preparation, the samples are exposed to a drying process which can greatly influence the morphological patterns, especially in the case of aggregate formation. The DLS results for a few hours aged Au-AgNSs and aged Au seeds are listed in Table 2.

Obviously, there is a bimodal particle size distribution of the final Au-AgNSs as well as of original Au seeds based on DLS intensity (Table 2). AuBhr and AuCitr revealed two types of similar mean particle sizes (around 40 and 10 nm), however, each of them in a different percentual content (Table 2).

In AuBhr_Ag+_Asc and AuCitr_Ag+_Asc, the majority of particle sizes (76% and 80%, respectively) seems to be concentrated around 67 and 65 nm, respectively. The standard deviation is of a rather high value, 38 nm for AuBhr_Ag+_Asc and 34 nm for AuCitr_Ag+_Asc. It indicates that the systems still develop in the moment of DLS measurements. The second maximum of particle size distributions, contributing by a quarter in AuBhr_Ag+_Asc and by 20% in AuCitr_Ag+_Asc, is centered at 11 and 10 nm, respectively. It should be reminded that scattering intensity of bigger particles is higher than that of smaller particles [60]. Therefore, the portion of smaller particles is underestimated and that of bigger particles is overestimated in these results.

Furthermore, hydrodynamic diameter is assessed in DLS measurements while solely metallic parts of NSs are visualized in TEM images. Indeed, the final Au-AgNSs travel across the solution together with several shells of small organic molecules and ions stemming from their syntheses (residues of citrates and/or acetone, (poly)borates together with dehydroascorbate and other potential ascorbic acid residues). Hence, it may lead to differences between particle sizes determined by DLS vs. TEM.

Considering the structures showed in Figure 1E and Figure 2C, the overall hydrodynamic diameter can be expected around 70 nm which correlates well with the values in Table 2. Furthermore, either the generated AgNPs are responsible for the second most abundant diameter (around 11 and/or 10 nm) detected by DLS, or Au seeds uncovered by Ag (less probable situation); their metallic parts (visualized by TEM in Figure 1and Figure 2) are well below these values. Based on DLS measurements, it becomes clear that an isolated small AgNP observed in the close vicinity of a gold seed with silver layer in representative TEM image of AuCitr_Ag+_Asc (Figure 2C), can stem from the drying process during TEM sample preparation. Similarly, many other small AgNPs in AuBhr_Ag+_Asc (Figure 1E) could be spread separately in the solution, moving thus independently of Au seeds as determined by DLS.

#### 3.2.2. Extinction Spectra of Au Seeds and of Final Au-AgNSs

According to UV-Vis spectroscopy, the LSPR maximum of AuBhr is located at 515 nm (curve a in Figure 3A). Upon Au-AgNSs formation, this maximum is shifted to 397 nm, LSPR is broadened with a shoulder around 350 nm and a significant red-tail (curve b in Figure 3A). This red-tail is most probably induced (i) by the absence of Ag layer around AuNPs (as evidenced in Figure 1), hence, Au NPs still partially contribute to the extinction spectrum, as well as, (ii) by the formation of AgNPs aggregates. The intensity of the LSPR maximum of AuBhr_Ag+_Asc is rather low (curve b in Figure 3A). This can be explained by the fact that smaller AgNPs, which manifest themselves by a broader LSPR and possess much lower scattering cross-section [60], are present in AuBhr_Ag+_Asc as it was evidenced by TEM and DLS.

The LSPR maximum of AuCitr is positioned at 525 nm (curve a in Figure 3B). Obviously, bigger starting AuNPs in AuCitr than in the case of AuBhr are present which is consistent with TEM imaging (Appendix A). The LSPR maximum of AuCitr_Ag+_Asc shifted to 401 nm and a slight red-tail can be distinguished (curve b in Figure 3B). Similarly, as in AuBhr_Ag+_Asc (curve b in Figure 3A), a shoulder at shorter wavelengths is detected in UV-Vis spectrum of AuCitr_Ag+_Asc (curve b in Figure 3B). We assume that it belongs to the transversal mode of LSPR [61] of asymmetric Au-AgNSs (as for instance visualized in Figure 2C). Simultaneously, the red-tail in UV-Vis spectra of AuCitr_Ag+_Asc can be attributed to a fraction of NPs being of such asymmetric shape and possessing longitudinal mode of LSPR [61]. Last, but not least, the band around 265 nm detected in AuCitr_Ag+_Asc can be attributed to ascorbate and/or dehydroascorbate.

Considering the results and discussed correlations and dependencies, distinctly different morphologies of the final Au-AgNSs and their related spectroscopic properties were achieved simply by using two different Au seed types (i.e., without any other additional surfactant).

### 3.3. Classical vs. Reverse Order of Reactants in Seed-Mediated Growth Procedure of Au-AgNSs

As mentioned in the introduction, only a few researchers have dealt with the effect of the order of reactants in the seed-mediated growth of Au-AgNSs so far. Moreover, their seeds were surface modified by surfactants which were intentionally added into the reaction mixture. Here, we selected aged forms of Au seeds (i.e., without any further surfactants added) and performed the synthetic procedures of the seed-mediated growth with the reverse order of reactants. In other words, Asc was mixed with Au seeds and, subsequently, AgNO_3_ was introduced. This can influence the characteristics of electrostatic double layer around AuNPs as depicted in Figure 2A,B.

While added Ag(I) can partially compensate the negatively charged envelope of Au seeds (Figure 2A), leading thus to a localization of silver ions in the close vicinity of AuNPs; the addition of Asc prior to Ag(I) (Figure 2B), can result in molecular exchange of negatively charged ions surrounding AuNPs, i.e., Asc directly adsorbs on Au seed surfaces. Consequently, an anisotropic growth (similarly as in [62]) can be more pronounced in the latter case.

As it was proofed by TEM imaging for AuBhr_Asc_Ag+ and AuCitr_Asc_Ag+ (Figure 4): irregularly shaped particles resulted from synthesis with the reverse order of reactants (Asc prior to AgNO_3_). Since we are not playing with the final concentration of AgNO_3_, neither with the presence and concentration of any additional ions (such as for instance halides as in refs [32,63,64]), the only parameter responsible for such changes in the final Au-AgNSs morphology must be the order of reactants. Therefore, the envisaged anisotropic growth due to the adsorption of Asc on Au seed surfaces was confirmed.

UV-Vis spectra of AuBhr_Asc_Ag+ and AuCitr_Asc_Ag+ were recorded and are directly compared with those of AuBhr_Ag+_Asc and AuCitr_Ag+_Asc in Figure 3A,B, respectively. The LSPR maximum of AuBhr_Asc_Ag+ (curve c in Figure 3A) is located around 398 nm and manifests itself by an increased intensity in comparison to that of AuBhr_Ag+_Asc (curve b in Figure 3A). The red-tail in AuBhr_Asc_Ag+ (curve c in Figure 3A) is observed, however, it is less pronounced than that of AuBhr_Ag+_Asc (discussed in the previous section).

The LSPR maximum of AuCitr_Asc_Ag+ (curve c in Figure 3B) is of a similar intensity as that of AuCitr_Ag+_Asc (curve b in Figure 3B). The position of LSPR maximum of AuCitr_Asc_Ag+ (curve c in Figure 3B) can be found around 396 nm which is 5-nm blue-shifted in comparison to AuCitr_Ag+_Asc (curve b in Figure 3B). Obviously, the red-tail and shoulder around 350 nm are detected in both systems, i.e., in AuCitr_Ag+_Asc (curve b in Figure 3B), as well as in AuCitr_Asc_Ag+ (curve c in Figure 3B), revealing thus the presence of asymmetric and prolongated NSs—in accordance with TEM images in Figure 4. Moreover, the band of ascorbate residues and/or dehydroascorbate (positioned at around 265 nm) evidenced their presence in the close vicinity of AuNPs even after several hours elapsed from the beginning of the synthesis. This confirms the idea of the direct adsorption of ascorbate and/or its residues on Au seed surfaces during the second step of seed-mediated growth.

Results of DLS measurements of AuBhr_Asc_Ag+ and AuCitr_Asc_Ag+ are summarized in Table 2 for the sake of a direct comparison with the systems discussed in previous section. Obviously, bimodal particle size distribution repeats in both systems similarly as it was in cases of AuBhr_Ag+_Asc and AuCitr_Ag+_Asc. However, while the mean size diameter of bigger particles of 55 ± 27 nm prevails (particle size distribution based on intensity) in AuBhr_Asc_Ag+; it is of 74 ± 39 nm in AuCitr_Asc_Ag+. The mean size diameter of the fraction of smaller particles is centered at 7 ± 3 nm and 13 ± 5 nm for AuBhr_Asc_Ag+ and AuCitr_Asc_Ag+, respectively (Table 2).

The differences in mean particle sizes of NSs generated by classical (Ag+ prior to Asc) vs. reverse (Asc prior to Ag+) order of reactants give evidence about the importance of this factor during the seed-mediated growth procedure. Moreover, it can be stated that the type of AuNPs has an even bigger impact on particle size distribution when reverse order of reactants is used. In other words, while bigger mean size particles are generated in AuBhr_Ag+_Asc than in AuBhr_Asc_Ag+; the trend is opposite in systems using AuCitr as seeds. Hence, the order of reactants influences the particle size distribution of the final Au-AgNSs. Indirectly, it confirmed the idea about Asc adsorption on AuNP surfaces and particularly, the role of (poly)borates vs. residues of citrates and/or acetone on Au seed surfaces: while the addition of Asc prior to Ag^+^ stabilizes AuBhr (by electro-steric stabilization); the same procedure destabilizes AuCitr (most probably due to a competitive adsorption of residues of citrates/acetone and Asc on AuNP surfaces).

Importantly, the pH values were determined for Au seeds (7.80 in AuBhr, whereas 6.02 in AuCitr), as well as, for the final Au-AgNSs (listed in Table 2). In both Au seeds solutions, Asc is in the dissociated form (because the pKa value is 4.2 ref. [65]), i.e., as ascorbate. While the final Au-AgNSs containing AuBhr seeds reached the pH values of 3.44 and 3.49; the systems containing AuCitr seeds revealed the pH values of 3.69 and 3.73. There is no wonder that the values were the same for each pair of Au-AgNSs within the experimental error (0.05) (Table 2) because the concentrations of all reactants are the same in classical and reverse order of reactants during the seed-mediated growth. Simultaneously, it was found by pH measurements that non-dissociated form of Asc should be present (if unconsumed) in the final Au-AgNSs.

### 3.4. Kinetics of Au-AgNSs Generation

UV-Vis spectroscopy is an ideal tool to investigate the process of Au-AgNSs generation due to the fact that Au seeds as well as the final Au-AgNSs possess surface plasmons. In Figure 5, extinction spectra in particular time intervals elapsed from the addition of the last reactant in the process of seed-mediated growth of Au-AgNSs are shown. The spectra are shifted from each other by a constant value in y-axis for the sake of their better mutual comparison. Due to the usage of stacked arrangement in Figure 5, the shape and changes in intensity of the extinction spectra can be immediately distinguished even by a naked eye.

Evidently, the process of reduction of Ag+ on Au seeds is taking place within the first minute and no further changes in surface plasmon extinctions are observed when the classical order of reactants is used (Figure 5A,B) and/or in the case of AuBhr_Asc_Ag+ (Figure 5C) because LSPR of Ag dominates the spectra. On the contrary, the process is slowed down in the reverse order of reactants (Asc prior to Ag+) when using AuCitr seeds (Figure 5D).

It can be concluded that the process of citrate residues replacement by Asc at AuCitr surface is much slower than the replacement of (poly)borates by Asc at AuBhr surface. Moreover, this corroborates the idea of a sort of destabilization of AuCitr seeds where residues of citrates and/or aceton are slowly replaced by Asc.

Long-term stability of Au-AgNSs was also investigated and discussed, but due to lack of space it is discussed in Appendix A.

### 3.5. SERS Application of Au-AgNSs

We assume that ascorbates, dehydroascorbates, (poly)borates, and oxidation products of citrates, which are available on surfaces of Au-AgNSs, can influence the detection of tetracationic TMPyP, especially the type of its SERS-spectral form. Four Au-AgNSs were investigated concerning their SERS-activity by using TMPyP: AuBhr_Ag+_Asc, AuBhr_Asc_Ag+, AuCitr_Ag+_Asc, AuCitr_Asc_Ag+.

Prior to the addition of TMPyP solution, the aggregation states of the four Au-AgNSs were first checked by recording their UV-Vis spectra. Then, after the addition of TMPyP, the UV-Vis spectra were remeasured to determine the spectral changes induced by the process of TMPyP and NSs interaction (see in Appendix A). Consequently, a degree of Au-AgNSs aggregation can be estimated and the extent of resonance of a particular system with the used laser wavelength can be evaluated. Generally, the more in resonance (laser wavelength with the band of adsorbate-NSs ensemble), the better SERS signal of the adsorbate should be obtained.

Considering our excitation laser wavelength (785 nm) and the extinction spectra of Au-AgNSs with TMPyP (Appendix A), more intensive SERS-spectral pattern of TMPyP can be envisaged in cases of Au-AgNSs prepared by using AuCitr seeds than in the Au-AgNSs exploiting AuBhr seeds. This correlates well with the assumption based on zeta potential values of these Au-AgNSs (in Appendix A): Au-AgNSs with AuCitr seeds are more prone to aggregation, which was also evidenced by UV-Vis spectra (Appendix A), than Au-AgNSs with AuBhr seeds.

In Figure 6, characteristic SERS spectra of each Au-AgNSs alone (black curves) and together with TMPyP (colored curves) are shown. Obviously, all four Au-AgNSs manifested themselves by relatively simple spectral features with the two well-developed most intensive bands: around 250 cm^−1^ and 1642 cm^−1^. While the former can be attributed to Ag-OOC- [66] of dehydroascorbate and/or ascorbate residues because it is found in all four Au-AgNSs (Figure 6A–D); the latter stems from deformation vibrations of water molecules. Other bands of weak intensity positioned at around 490, 600, 1050, and 1420 cm^−1^ can be found in SERS-spectra of Au-AgNSs (black curves in Figure 3A–D). Based on peak positions, they could be assigned to dehydroascorbate and/or ascorbate residues present at the surface of Au-AgNSs.

The addition of TMPyP to Au-AgNSs results in changes of SERS-spectral pattern: the peak at 250 cm^−1^ enormously increased in intensity and many characteristic peaks of TMPyP appeared (in Figure 6A–D). Particularly, the SERS pattern of free-base TMPyP (as clearly determined in [40]) can be unambiguously distinguished by: (i) doublet positioned at around 336 and 406 cm^−1^, (ii) peak located at approx. 1006 cm^−1^, (iii) doublet at 1336 and 1366 cm^−1^. The only missing characteristic peak of free-base TMPyP SERS-spectral form is that positioned at 1562 cm^−1^ which is most probably shifted to 1548 and/or 1554 cm^−1^ in our cases (blue and red curves in Figure 6A–D, respectively).

The final TMPyP concentration was intentionally chosen to reach the value of 1 × 10^−8^ M because it is well below the calculated monolayer coverage of TMPyP on Au-AgNSs (more than one order of magnitude, for detailed calculation see Appendix A). Therefore, all molecules of TMPyP should be able to interact directly with Au-AgNSs.

We assume that free-base TMPyP is detected due to the presence of ascorbic acid residues (manifesting themselves by the intensive peak at 250 cm^−1^) which serve as a type of molecular spacer—similarly as mercaptoacids in [37] and gluconic or glucuronic acids in [38]. It can be thus concluded that free-base form of TMPyP in submicromolar concentrations (10 nM) is detected in all four Au-AgNSs regardless the type of Au seeds, as well as order of reactants in the second step of the seed-mediated growth procedure (compare SERS spectral patterns in blue and red curves in Figure 6A–D, respectively). This can be of interest in bio-analytical detection of porphyrins.

Based on negative zeta potential values of Au-AgNSs (listed in Appendix A), aggregation caused by TMPyP addition (verified by UV-Vis spectra—Appendix A), and SERS detection of free-base TMPyP (Figure 6A–D), we propose an ionic interaction between tetracationic TMPyP and negatively charged dehydroascorbates and ascorbic acid residues present at Au-AgNSs surface. For the sake of a direct comparison, TMPyP was added into AgBhr colloid (separately prepared and aged for 9 months—experimental details are in Appendix A) where characteristic SERS spectral peaks of metalation markers of Ag(I)-TMPyP were detected (398, 1342, 1546 cm^−1^)—see Appendix A. Moreover, TMPyP SERS-spectral pattern was checked using Au seeds prepared by borohydride and/or citrate reduction in order to compare with SERS features of TMPyP on Au-Ag NSs (in Appendix A). Obviously, the metalation markers of Ag-TMPyP cannot be detected since no Ag is present in Au seeds. Importantly, the SERS signal of TMPyP on Au-AgNSs is better than that obtained by using Au seeds alone.

As for the intensity of TMPyP SERS-signal (Figure 6A–D), it correlates well with the trend proposed by interpretation of their UV-Vis spectra (Appendix A): more intensive in Au-AgNSs containing AuCitr as seeds than in Au-AgNSs exploiting AuBhr seeds. There are slight intensity changes of TMPyP signal (based on the intensity of the peak located, for instance, at 1190 cm^−1^) in Au-AgNSs containing AuCitr seeds when the order of reactants changed: TMPyP signal is a little bit more intensive in AuCitr_Ag+_Asc than in AuCitr_Asc_Ag+ (Figure 6C,D). Similarly, SERS signal of TMPyP is more pronounced in AuBhr_ Ag+_Asc than in AuBhr_Asc_Ag+ (Figure 6A,B). It can be explained by different surface chemistry and characters of electric double layer surrounding Au-AgNSs which was induced by the order of reactants in the second step of the seed-mediated growth as it was schematically depicted in Figure 2 and discussed in previous sections.

The enormous increase of the SERS intensity of the peak attributed to dehydroascorbate and/or ascorbate residues (located at 250 cm^−1^) induced by TMPyP addition to Au-AgNSs, can be explained by the aggregation process caused by the addition of tetracationic molecules of TMPyP to negatively charged Au-AgNSs (i.e., revealing negative zeta potential values as discussed in section of Appendix A when long-term stability discussed). Consequently, the LSPR peak is red-shifted and comes more in resonance with the excitation laser wavelength, leading thus to the observed SERS intensity increase. Interestingly, there are obvious differences in systems with classical vs. reverse order of reactants: the peak of 250 cm^−1^ is more enhanced in Au-AgNSs containing AuBhr as seeds when the order Asc prior to Ag+ (i.e., the reverse order of reactants) used; while its enhancement is higher in AuCitr_Ag+_Asc than in AuCitr_Asc_Ag+ (Appendix A). Taking into account the assumption depicted in Figure 2, Asc can replace weakly adsorbed (poly)borates and citrates, however, the former possibly more easily than the latter (as evidenced by the kinetics of Au-AgNSs formation in the previous section), leading thus to the observed differences in SERS signal stemming from residues of Asc and dehydroascorbate.

Based on SERS spectra, it can be summed up that both factors, Au seed types and the order of reactants in the second step of the seed-mediated growth, play a role in quantitative, but not qualitative SERS-spectral behavior of the final Au-AgNSs when these are allowed to interact with a cationic adsorbate via ionic interaction. Importantly, the free base form of TMPyP is detected in all four selected Au-AgNSs types, therefore reporting a well-developed envelope of organic molecules around each Au-AgNSs which prevents the metalation to occur.

## 4. Conclusions

Core-shell Au-AgNSs were prepared by a seed-meditated growth exploiting several different types of Au seeds and varying the order of AgNO_3_ and Asc addition (classical: Ag+ followed by Asc, and/or reverse order: Asc prior to Ag+). It was revealed that the type of Au seeds, as well as the order of reactants in the second step of the seed-mediated growth procedure, both had a great impact on the final Au-AgNSs characteristics as determined by TEM imaging, UV-Vis spectroscopy, DLS, and zeta potential measurements. While AuNPs enveloped by a bunch of tiny AgNPs in the case of borohydride-reduced Au seeds were observed; a compact Ag layer around AuNPs was detected in the case of Au-AgNSs when citrate-reduced Au seeds were employed. Moreover, an envelope of organic molecules (dehydroascorbate and Asc residues) is detected on each Au-AgNSs as determined by spectroscopic measurements (SERS, UV-Vis) as well as based on negative zeta potential values. When the selected as-prepared Au-AgNSs applied in SERS detection of the model tetracationic porphyrin, 5,10,15,20-tetrakis(1-methyl-4-pyridyl)21H,23H-porphine (TMPyP), unperturbed (free-base) form of TMPyP was detected in submicromolar concentrations (10 nM), thus bioanalytical application of these Au-AgNSs can be envisaged.

## Data Availability

The data is included in the main text and/or the Appendix A.

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
