# Peer review of "Distinctly Different Morphologies of Bimetallic Au-Ag Nanostructures and Their Application in Submicromolar SERS-Detection of Free Base Porphyrin"

_nanomaterials, 2021, doi:10.3390/nano11092185_

Round 1
Reviewer 1 Report
The author revised the manuscript according to the reviewer’s suggestions. The added TEM images provide the different morphologies of bimetallic Au-Ag nanoparticles. Overall, this study constitutes a large body of work and the informative results include. However, I am afraid that it is difficult to follow the conclusive result in the manuscript since the discussion tends to be divergent. To improve the quality of the article, even now, several points should be addressed before the acceptance.
1.The sentence “nanosunflower” would mislead the reader. The dark core and light shell in the TEM image originate from Ag-Au core-shell NPs, and they are not specific morphology. In general, “Flower-like nanoparticles” have nanometer-sized protrusions on the surface, resulting in SERS enhancement.
2.Types of Au seeds and added order of reactants in seed-mediated growth provide different morphologies of bimetallic Au-Ag nanoparticles. These findings are interesting. To make it easier to understand how we can obtain these different NPs, I suggest that the author add a schematic figure showing the relation between the synthetic procedure and the class of NPs obtained in this study. From what I understand, the classes may be divided into the following: (class 1) a compact Ag layer around AuNPs (i.e., Ag-Au spherical NPs with core-shell structure, Fig. 2), (class 2) irregular shaped Au NPs surrounded by small AgNPs, Fig. 1A, (3) the mixture of Au NPs and Ag NPs, Fig. 1F).
3.The AuBH-fr seeds give the NPs of class 2 described in comment 2, but the AuCitr-fr seeds give the class 1 NPs. The explanation for this reason will be helpful for readers on how we control the morphology of bi-metallic NPs by the seed types. From Fig. S1, seeds' size and morphology differ between AuBH-fr seeds and AuCitr-fr seeds. Does such different morphology provide different morphological bi-metallic NPs? Citric acids are also known as a metal-chelate agents. Does the chelate ability with Ag ion cause more adsorption of Ag ions onto citrate -Au NPs resulting in Ag-Au spherical NPs with core-shell structure?
4.The kinetics data of Au-AgNSs generation by monitoring the time-dependent UV-Vis spectra are important to understand the formation mechanism. However, the data causes me some confusion in understanding the meaning. Figs 5A, 5B, 5C show the absorbance decreases with the same absorption shape as the reaction (the process of reducing Ag+ on Au seeds ) proceeds. Such change is often observed in the sedimentation of NPs. The morphological evolution of NPs should provide different absorption shapes. The absorption peak appears at around 400 nm, originating from plasmonic Ag NPs in Figs 5A, 5B, 5C. Why does the process of reduction of Ag+ on Au seeds cause the absorption peaks as shown in these figures. This observation also confuses me.
Author Response
We thank to referee 1 for her/his suggestions and comments. We address to each point in the following text written by blue colour.
We would like to pinpoint that the position of the cited lines in our revised manuscript are related to the text showing all revisions (the mode “all revisions visible”).
Comments and Suggestions for Authors
The author revised the manuscript according to the reviewer’s suggestions. The added TEM images provide the different morphologies of bimetallic Au-Ag nanoparticles. Overall, this study constitutes a large body of work and the informative results include. However, I am afraid that it is difficult to follow the conclusive result in the manuscript since the discussion tends to be divergent. To improve the quality of the article, even now, several points should be addressed before the acceptance.
1.The sentence “nanosunflower” would mislead the reader. The dark core and light shell in the TEM image originate from Ag-Au core-shell NPs, and they are not specific morphology. In general, “Flower-like nanoparticles” have nanometer-sized protrusions on the surface, resulting in SERS enhancement.
Our response 1:
Prompted by this referee comment, we removed the word “nanosunflower” from the title of the manuscript as well as from Abstract – lines 2 and 9, respectively.
We have also removed this statement in brackets in lines 111-112 „(the flower-like image is evoked due to their dark core and light shell in TEM images)” and included the following sentence in lines 113-115: „In general, flower-like nanoparticles possess nanometer-sized intersticies on the surface which results in SERS enhancement.“
We removed the sentence - lines 327-329: „This resembles flowers and therefore we name them as flower-like NPs of different morphologies.“
The following sentence has been added - lines 380-381: “This resembles flower-like particles with many nanometer-sized intersticies and SERS enhancement can be envisaged.“
2.Types of Au seeds and added order of reactants in seed-mediated growth provide different morphologies of bimetallic Au-Ag nanoparticles. These findings are interesting. To make it easier to understand how we can obtain these different NPs, I suggest that the author add a schematic figure showing the relation between the synthetic procedure and the class of NPs obtained in this study. From what I understand, the classes may be divided into the following: (class 1) a compact Ag layer around AuNPs (i.e., Ag-Au spherical NPs with core-shell structure, Fig. 2), (class 2) irregular shaped Au NPs surrounded by small AgNPs, Fig. 1A, (3) the mixture of Au NPs and Ag NPs, Fig. 1F).
Our response 2:
We added subchapter „3.1.3 Reason for different morphologies of Au-AgNSs“ into our revised version of the manuscript – line 423, in order to improve the clarity of the text.
Moreover, we prepared Scheme 1 which includes three different types of morphologies observed on TEM images. Accordingly, the previous Scheme 1 was renumbered to Scheme 2 and both are cited in the main text under these new numbers.
We discussed the reason in more details as it can be read in lines 424-427: „It can be summed up that three different types of morphologies, schematically depicted in Scheme 1, were obtained simply by changing Au seeds. While type I represents a mixture of Au and Ag NPs, type II resembles flower-like particles with nanometer-sized intersticies, and type III is characteristic by a consistent layer of Ag around Au (Scheme 1).“
Lines 431-432 were also added: „Citrates are known as chelating agents of silver cations [21] [39].“
Lines 434-435 were added: „Therefore, it can be deduced that the creation of type III is caused by the chelating effect of citrate ions on silver cations.“
Lines 447-453 were added: „Morphologies of type II (Scheme 1), observed when AuBH-fr seeds employed (Figures 1A,B), can be caused by the presence of still reactive borohydride on Au seeds. Indeed, it leads to the direct reduction of Ag+ in the very close vicinity of freshly prepared Au seeds.
On the contrary, morphologies of type I (Scheme 1), which are characteristic for AuBhr_Ag_Asc and AuBOH-fr_Ag_Asc (Figures 1C-1F), are most probably induced by the presence of either polyborates, or borates which hinder the direct deposition of Ag+ on Au seed surfaces (i.e., poly/borates are not chelating agents of silver cations).“
3.The AuBH-fr seeds give the NPs of class 2 described in comment 2, but the AuCitr-fr seeds give the class 1 NPs. The explanation for this reason will be helpful for readers on how we control the morphology of bi-metallic NPs by the seed types. From Fig. S1, seeds' size and morphology differ between AuBH-fr seeds and AuCitr-fr seeds. Does such different morphology provide different morphological bi-metallic NPs? Citric acids are also known as a metal-chelate agents. Does the chelate ability with Ag ion cause more adsorption of Ag ions onto citrate -Au NPs resulting in Ag-Au spherical NPs with core-shell structure?
Our response 3:
Due to the fact that new Scheme 1 and new subchapter (3.1.3) were included in the revised version of our manuscript, as stated in our response 2 above, it could be repeated as a reaction on this referee comment here.
Indeed, we assume that citrates bind Ag+ more easily than poly/borates; the reason for that is most probably citrates chelating ability. Therefore, the morphology of the final Au-AgNSs differs significantly as schematically depicted in Scheme 1: from randomly dispersed mixtures of Au and Ag NPs, through an uneven layer of Ag NPs around Au seeds (flower-like morphology), to a compact layer of Ag on Au core.
The shape of Au seeds might influence the final morphology of Au-AgNSs till some extent. However, it is well known that in both cases, AuBhr and AuCitr, a fraction of irregularly shaped nanoparticles is usually formed besides nearly spherical ones (which represent the majority of NPs shape). It is also shown in Figure SI-1. That is why we rather propose that differences in surface chemistry of Au seeds are responsible for the differences in morphology in the sense of Scheme 1, i.e., compact layer vs. uneven layer of Ag on Au seeds vs. mixture of Ag and Au NPs. This assumption is further supported by the comparison of freshly prepared vs. aged Au seeds.
4.The kinetics data of Au-AgNSs generation by monitoring the time-dependent UV-Vis spectra are important to understand the formation mechanism. However, the data causes me some confusion in understanding the meaning. Figs 5A, 5B, 5C show the absorbance decreases with the same absorption shape as the reaction (the process of reducing Ag+ on Au seeds ) proceeds. Such change is often observed in the sedimentation of NPs. The morphological evolution of NPs should provide different absorption shapes. The absorption peak appears at around 400 nm, originating from plasmonic Ag NPs in Figs 5A, 5B, 5C. Why does the process of reduction of Ag+ on Au seeds cause the absorption peaks as shown in these figures. This observation also confuses me.
Our response 4:
Possibly, we were too brief in our introduction of Figure 5 in the main text. This was then confusing for the referee and may be confusing for any potential reader.
Therefore, the sentences in lines 702-705 were added: “The spectra are shifted from each other by a constant value in y-axis for the sake of their better mutual comparison. Due to the usage of stacked arrangement in Figure 5, the shape and changes in intensity of the extinction spectra can be immediately distinguished even by a naked eye.“
Here (visible in pdf file), we are showing overlaid extinction spectra of AuBhr_Ag_Asc (left) and AuBhr_Asc_Ag (right) as examples of another possibility of their presentation (taken from our working files). However, this presentation is not clear enough, in our opinion (please, see page 3 of the pdf file attached). Therefore, we decided to use stacked spectra in Figure 5.
Furthermore, we would like to remind that localized surface plasmon resonance (LSPR) of Ag NPs is located for spherical particles around 400 nm, while LSPR of Au NPs (spherical) around 520 nm. It is also shown in Figure 3 and mentioned in the main text. Since we are using only 0.5 mL of Au seeds and 13 mL is the final volume of the reaction mixture, the intensity of LSPR peak is determined by generated Ag NPs and/or Ag layers on Au seeds. In other words, we included sentence in lines 714-715 „ because LSPR of Ag dominates the spectra“.
For the sake of clarity we added „(Asc prior to Ag+)“ in line 716, and „by Asc“ in lines 717 and 718.
We would like to thank to this referee once again for her/his time and hope that the revised version is now much more comprehensible for all potential readers.
Reviewer 2 Report
I think the authors have revised the manuscript according the previous comments; however, English editing is required. Especially an obvious errors in the figure captions where the figure's number is wrong. Please send the manuscript to the English editing service.
Author Response
We thank to referee 2 for her/his effort to improve our manuscript.
There is nothing else than English editing required. We repeat that our linguist read the manuscript carefully and agreed with the last version we have resubmitted. Possibly, she did not checked captions of each figure. Currently, it is revised as can be seen in lines: 388, 419, 512-515, 576, 593, 710, 766-767.
Interestingly, the other referees did not require any extensive editing of English anymore.
Reviewer 3 Report
The corrected version of the manuscript is improved accordingly with the 3 referees' comments. In my opinion now the manuscript is suitable for publication.
Some minor style and grammar errors are still present but they can be corrected in a final stage.
Author Response
We thank to referee 3 for her/his time in this second round of revisions.
Round 2
Reviewer 1 Report
The authors have addressed the issues in a due way. I suggest its acceptance.
This manuscript is a resubmission of an earlier submission. The following is a list of the peer review reports and author responses from that submission.
Round 1
Reviewer 1 Report
The authors report the synthesis of core-shell Au-Ag NSs and their applications for SERS.
The effect of seed Au NPs, which are prepared by several methods, on the size of the morphology of Au-Ag NSs was examined. This viewpoint is interesting, but the resultant Au-Ag NSs are undefined morphology, and the size differences are not apparent. The discussion is difficult. The aging for a very long time of 7 months makes it more difficult to obtain the critical factor. At least, TEM images of all seeds are needed. I don't think that the current set of data are enough to draw the conclusions.
Reviewer 2 Report
The authors used Ag-Au core-shell NPs as SERS for detecting porphyrin. Such kind of research topics have been studies for several years by the corresponding author. Without comparing the author's current results with other previous published works, it is not easy to draw researchers' interests reading the manuscript. Several suggestions:
- The title is " Distinctly different morphologies of bimetallic Au-Ag nanosunflowers"; however, we only can see one figure (figure 1) as morphology of NPs. Actually, Figure 1 could hardly tell the different NPs fabricated using different seed solutions. If the authors still want to keep the title alive, they must show more TEM images.
- The Abstract: It is not easy to understand which part is the current work and which part is the previous work. Also many sentences are too long to understand.
- The introduction : In the line 53, the authors already try to tell the readers their current work; however, in the line 71, they again reviewed the previous works. It is not a common writing style for scientific paper. Also, in the introduction part, the authors should review different NPs for detecting porphyrin or using same NPs detecting different type of porphtrin. The reason why we would like to use prophyrin as SERS target molecules should also be discussed.
- The authors compare the effects of seed solution, freshly prepared and aged 7 month. I am not sure why they choice 7 month. If the mediation of seed solution has big impact on the growth of Ag-Au NPs, they should meditate the seed solution for different months and compare the results. Also they used different reducing agent. I believe the readers would like to see the TEM images of Au NPs before capping the Ag using different Au seed solutions. Or the change of Absorption peak of Au seed solutions during meditation. Without those discussions, it is not easy to tell the impact of the current manuscript.
- The headline in the manuscript (results and discussion part) could not help the readers to clearly understand the manuscript. For examples, "3.1 Direct impact of Au seed type on characteristics of final Au-AgNSs", and the following discussion is not showing the results, instate, the authors again reviewed previous works. I believe line 230-254 should move to the introduction part.
- In the headline "3.1.1 Comparison of AuBhr-7m and AuCitr-7m serving as seeds", actually, the authors only compared two different methods fabricating NPs, then, they should only show Figure 1 (A) and (B). Also they should show those fabricated NPs in different magnifications.
- The authors should added more headline to discuss the effect of "freshly prepared Au seed layers"
- The SERS results should also show the prophyrin signal using the freshly prepared Au seed layers.
Reviewer 3 Report
this manuscript is a good technical paper on the preparation of AuAg nanoparticles. the authors investigated extensively several parameters used during the synthesis, studying the morphology and the optical response of the obtained particles.
the manuscript is rather long and for this reason hard to be read and followed, anyway the large amount of information reported justify (partially) the length.
I think that it can be published once some improvements will be done
in particular: figure 1 needs a scale bar with a value.; table 2 and table 3 should report clearly which is the size of Au and Ag particles; figure, to me should be divided in 2 separated figures discussed better, a legend within the graph should be also useful.
the authors could consider to arrange in a better way the discussion also in order to reduce a little bit the length of the paper.